# Innovative Solutions for Worn Fingerprints: A Comparative Analysis of Traditional Fingerprint Impression and 3D Printing

**DOI:** 10.3390/s24082627

**Published:** 2024-04-20

**Authors:** Wenhui Mao, Yadong Zhao, Petro Pavlenko, Yihan Chen, Xuezhi Shi

**Affiliations:** 1School of Food and Pharmacy, Zhejiang Ocean University, No. 1 Haida South Road, Dinghai District, Zhoushan 316022, China; maowenhui@zjou.edu.cn (W.M.); yadong@kth.se (Y.Z.); 2School of Marine Engineering Equipment, Zhejiang Ocean University, No. 1 Haida South Road, Dinghai District, Zhoushan 316022, China; petrpavlenko@zjou.edu.cn (P.P.); chenyihan@zjou.edu.cn (Y.C.)

**Keywords:** fingerprint, fingerprint impression technology, 3D printing technology, backup, unlock

## Abstract

Fingerprint recognition systems have achieved widespread integration into various technological devices, including cell phones, computers, door locks, and time attendance machines. Nevertheless, individuals with worn fingerprints encounter challenges when attempting to unlock original fingerprint systems, which results in disruptions to their daily activities. This study explores two distinct methods for fingerprint backup: traditional fingerprint impression and 3D printing technologies. Unlocking tests were conducted on commonly available optical fingerprint lock-equipped cell phones to assess the efficacy of these methods, particularly in unlocking with worn fingerprints. The research findings indicated that the traditional fingerprint impression method exhibited high fidelity in reproducing fingerprint patterns, achieving an impressive unlocking success rate of 97.8% for imprinting unworn fingerprints. However, when dealing with worn fingerprints, the traditional fingerprint impression technique showed a reduced unlocking success rate, progressively decreasing with increasing degrees of finger wear. In contrast, 3D-printed backup fingerprints, with image processing and optimization of ridge height, mitigated the impact of fingerprint wear on the unlocking capability, resulting in an unlocking success rate of 84.4% or higher. Thus, the utilization of 3D printing technology proves advantageous for individuals with severely worn or incomplete fingerprints, providing a viable solution for unforeseen circumstances.

## 1. Introduction

Fingerprints, characterized as unique biometric features of the human body [1], exhibit distinctiveness and immutability over time [2]. With intricately complex patterns, fingerprints pose a formidable challenge to forging, which makes them indispensable for various applications such as document signing and attendance tracking. In the twenty-first century, fingerprints have evolved into a pivotal tool for enhancing information security. As fingerprint technology continues to mature, its application scope has significantly broadened, finding widespread utility in fields like identity verification [3], medical applications [4,5], and forensic identification [6,7]. However, certain demographic groups, including individuals with disabilities, those engaged in manual labor, or professionals in specialized occupations such as swimmers and researchers exposed to chemicals, often confront issues like partial or worn fingerprints. The disappearance of fingerprints presents substantial challenges for these individuals, affecting activities such as unlocking smartphones [8], visa processing, attendance tracking, and banking transactions. While superficially damaged fingerprints can be restored [9], the recovery period of fingerprint skin is relatively extensive [10]. Therefore, backing up personal fingerprints and fabricating fingerprint membranes [11] become crucial to address unforeseen circumstances.

With the rapid advancement of technology, the design and functionalities of smartphones are continually evolving. A variety of biometric characteristics, including face, iris, fingerprint, keystroke, signature, and voice, have been utilized [12,13]. Each biometric trait possesses its own set of strengths and weaknesses, and their selection depends on the specific application. Among these advancements, in-display fingerprint scanning technology has emerged as a recent solution in the mobile industry [14,15]. This technology utilizes the screen as a fingerprint recognition sensor, employing optical [16] or ultrasonic [17,18] methods to collect fingerprint information. It then compares this information with pre-stored fingerprint data to achieve fingerprint unlocking. The principle behind optical fingerprint recognition involves the refraction and reflection of light. When a user places their finger in the unlocking area, the region is illuminated, causing different angles and intensities of reflected light to occur on the “ridge lines” and “valley lines” of the fingerprint (Figure 1). These reflections pass through the pixel gaps of the screen and are received by an optical sensor beneath the screen, enabling fingerprint identification [19]. In contrast, ultrasonic fingerprint recognition relies on the contact of ultrasonic waves with the “ridge lines” and “valley lines” of the fingerprint (Figure 1). Differences in absorption, penetration, and reflection levels produce echoes of varying energy, which are then detected by the sensor, allowing for the determination of the specific form of the fingerprint [18]. The distinction between the two methods lies in the fact that optical fingerprint recognition captures 2D fingerprint images, while ultrasonic fingerprint recognition senses the 3D morphology of the fingerprint. Optical and ultrasonic in-display fingerprint locks represent the mainstream solutions currently available in the market, widely adopted by numerous renowned smartphone brands globally, including Apple, Samsung, Huawei, Xiaomi, and others [20].

Fingerprint impression is the most widely used fingerprint preparation method [21]. This procedure entails directly molding one’s finger on modeling hot-melt glue, enabling the hot-melt glue to solidify and cool, ultimately yielding a fingerprint model. Subsequently, a fingerprint membrane is cast using this model. Generally, two primary types of fingerprint membranes are available: regular silicone fingerprint membranes and capacitive–conductive silicone fingerprint membranes.

Numerous researchers have conducted experiments using the fingerprint impression technique to circumvent smartphone fingerprint locks. The research outcomes indicated that fingerprint membranes produced through fingerprint impression could successfully unlock smartphones. The ease of breaching smartphone security varied among different brands and models. For some smartphones, creating a regular silicone fingerprint model was sufficient, while others necessitated the fabrication of capacitive–conductive silicone fingerprint models. Despite its noteworthy effectiveness in backing up fingerprints, the fingerprint impression technology demands a certain level of molding skill. In comparison to genuine fingerprints, those prepared through fingerprint impression may exhibit issues such as “blank” or “broken” features, irregular edges, uneven surfaces, and indistinct secondary features.

Since the outbreak of the COVID-19 pandemic in 2019, there has been an urgent demand for non-contact collection and more precise and efficient fingerprint backup techniques. In recent years, 3D printing, an emerging digital manufacturing technology, has rapidly advanced. Stereo-lithography Apparatus (SLA) technology, the earliest commercialized form of 3D printing, demonstrates higher part accuracy compared to other 3D printing techniques, achieving a precision within 10 μm. Research indicates that the spatial distance between the ridge and the valley lines of human fingerprints is approximately 60 μm [22]. Therefore, the precision of SLA printing is sufficient to meet the requirements for preparing fingerprint molds, providing an alternative means of obtaining fingerprint membranes. This entails the utilization of only a fingerprint photograph, processing software, and a 3D printer to unlock a smartphone. Consequently, several researchers have explored the use of the SLA printing technology for fingerprint backup. Maro et al. [23] utilized SLA printing to fabricate artificial fingerprint membranes, successfully unlocking the fingerprint lock systems of iPhone 6, iPhone 8, Samsung Galaxy S8, Meizu M5s smartphones, iPad Air 2, and Schenker XMG A507 laptop. Arora et al. [24] employed 3D printing technology to design and manufacture a wearable fingerprint device that interacts with fingerprint reader panels, replicating the behavior of a normal finger and facilitating various operational settings. However, the existing studies predominantly concentrated on backing up unworn fingerprints, a task for which the traditional fingerprint impression technique can yield similar results with pristine fingerprints. The advantage of employing 3D printing technology for fingerprint backup lies in its capability to address the challenges faced by traditional fingerprint impression methods in accurately replicating fingerprints from unclean or worn fingers through image processing. Nevertheless, research on obtaining, processing, and recognizing fingerprint images in a damaged state using 3D printing technology is presently limited.

In this study, we addressed fingerprints with varying degrees of wear by employing both traditional fingerprint impression techniques and the SLA printing technology to create fingerprint molds and then conducted unlocking tests on several widely available smartphones equipped with in-display fingerprint recognition systems, recording the unlock success rates. Through experimental research, the unlocking success rates of these two fingerprint backup methods for fingerprints with different levels of wear were compared. Additionally, the impact of factors such as fingerprint image processing and ridge height settings on SLA-printed fingerprint molds were also investigated. The objective of this study is to provide a more comprehensive understanding of fingerprint backup technologies, aiming to address the challenges faced by specific user groups who encounter difficulties in normal fingerprint unlocking. Furthermore, this research contributes insights that may inform the security considerations of future fingerprint unlocking systems.

## 2. Materials and Methods

### 2.1. Experimental Materials

The materials utilized in this study encompassed those required for fingerprint collection, the preparation of fingerprint molds, and casting fingerprint membranes. The materials for fingerprint collection included glass slides, brushes, and 3000-grit black magnetic powder. The molding material for traditional fingerprint impression was hot-melt glue, while the material for the SLA printing molds was photosensitive resin. The materials for casting fingerprint membranes consisted of conductive silicone and a curing agent. The device used for capturing the fingerprint images was the Huawei Mate 30 Pro smartphone, equipped with a dual 40-megapixel high-definition main camera. Vaseline served as the mold and fingerprint membrane release agent. To simulate different degrees of fingerprint wear, sandpaper with a grit of 400 was employed to polish the fingerprints.

The smartphones selected for fingerprint membrane unlocking tests included commonly available models featuring in-display fingerprint locks: Huawei Mate 30 Pro, Huawei Mate 40 Pro, Xiaomi 10, Honor 10, and OPPO Reno 6. The SLA printing device utilized was the JGMaker G3 3D printer from Aurora Technology Co., Ltd. in Shenzhen, China.

### 2.2. Experimental Methods

This study employed two methods, fingerprint impression and SLA printing, for the preparation of fingerprint molds. The workflow for backing up the fingerprint membranes using the fingerprint impression method is illustrated in Figure 2 and can be delineated into three main steps: fingerprint molding, casting the fingerprint membrane, and conducting fingerprint-lock unlocking tests. The process began by directly molding a subject’s finger on a plastic material, followed by creating a mold of the fingerprint and subsequently casting the fingerprint membrane. The final step involved testing the fingerprint lock and calculating the success rate.

The workflow for backing up fingerprint membranes using the SLA printing method is illustrated in Figure 3 and encompassed the following 7 steps: fingerprint extraction, image processing, 3D model creation, SLA printing of the fingerprint mold, post-treatment, casting the fingerprint membrane, and conducting fingerprint-lock unlocking tests. The process initiated with the collection and extraction of fingerprints using the “powder dusting” method. The obtained fingerprint images underwent processing, and a 3D model of the fingerprint was generated. Subsequently, the SLA printing method was employed to fabricate the fingerprint mold. Upon completion of the printing process, post-processing was conducted on the fingerprint mold. Finally, the fingerprint membrane was cast, and the unlock success rate was calculated through fingerprint-lock testing. The detailed steps can be viewed in Appendix A.

## 3. Experiments for Fingerprint Backup

### 3.1. Fingerprint Backup Experiments Using the Fingerprint Impression Technology

To begin, we enrolled the fingerprints of three fingers (thumb, index finger, and middle finger), each in an unworn condition, and stored them in the fingerprint lock system. To investigate the impact of fingerprint wear on the unlock success rate of fingerprint membranes, sandpaper was used to polish the fingerprints of each finger 0, 10, 20, and 30 times. Utilizing the fingerprint impression technology, corresponding fingerprint molds were crafted (Figure 4a–d). To minimize errors during the imprinting process, each finger was independently pressed three times, creating three molds. Subsequently, the respective fingerprint membranes were prepared, as depicted in Figure 4e,f. In Figure 4, it is evident that as the number of polishing times increased, the degree of fingerprint wear intensified, resulting in a continuous reduction in ridge height. This reduction led to increasingly unclear ridge patterns in the cast fingerprint membranes.

### 3.2. Fingerprint Backup Experiment Using SLA Printing

The fingerprints of three fingers (thumb, index finger, and middle finger) after 30 polishing times were selected for SLA preparation. Initially, finger impressions were left on glass surfaces [25], and the fingerprints were collected using the “powder dusting” method on glass slides [26]. Subsequently, fingerprint images were captured using a camera (Figure 5a). The images underwent processing using Photoshop CS6 software and image enhancement algorithms (Figure 5b). Following this, ZBrush 2022 software was employed to transform the 2D fingerprint images into 3D fingerprint models (Figure 5c), with ridge heights set at 30, 60, 90, and 120 μm. After the creation of the 3D fingerprint models, they were imported into ChiTuBox v1.4.0 slicing software for slicing, and the SLA 3D printer was then used to print the fingerprint molds (Figure 5d) [27]. After the completion of the printing process, the fingerprint molds were soaked in anhydrous ethanol for 20 min to remove the surface photosensitive resin. Subsequently, UV light was applied for 10 min to post-cure the fingerprints on the molds. Additionally, to assess the impact of image processing on fingerprint unlocking efficacy, comparative experiments were conducted using untreated (Figure 5a) and processed (Figure 5b) fingerprint images to establish 3D fingerprint models.

### 3.3. Unlocking Test 

For both fingerprint molds prepared using fingerprint impression and SLA technology, a thin layer of Vaseline was applied to the fingerprint molds before casting to facilitate demolding. Subsequently, conductive silicone was applied to the fingerprint molds (Figure 6a). To expedite the curing of conductive silicone, a curing agent was added to help silicone dry quickly. Once conductive silicone had solidified, the fingerprint membranes were peeled off from the molds (Figure 6b). Each mold was cast three times, and using the cast fingerprint membranes, unlocking tests were conducted on five smartphones (Huawei Mate 30 Pro, Huawei Mate 40 Pro, Xiaomi 10, Honor 10, and OPPO Reno 6), as depicted in Figure 6c. 

In this study, we investigated the performance of three fingers (thumb, index finger, and middle finger) entered into the fingerprint unlocking system. Each finger was imprinted with three fingerprints under each test condition, and three fingerprint films were made for each of the fingerprints obtained. The fingerprint films obtained were subjected to five unlocking tests on each phone. The unlock success rate was determined by dividing the sum of the number of successful fingerprint unlocking attempts by the number of tests, performed one hundred and thirty-five times.

## 4. Results and Discussion

### 4.1. Impact of Finger Wear on the Unlocking Capability of Fingerprint Membranes Produced by Fingerprint Impression

The results of the unlocking tests for fingerprint membranes prepared using fingerprint impression are depicted in Figure 7. The experimental findings indicated that it was difficult for the unlock success rates of all fingerprint membranes to reach 100.0%. This discrepancy might be attributed to the misalignment of the fingerprint membranes during testing, leading to inaccurate placement in the fingerprint recognition area, thereby hindering accurate identification. Unpolished fingerprints exhibited higher unlock success rates, ranging from 91.1% to 97.8%. As the number of polishing times increased, the unlock success rates of the cast fingerprint membranes steadily decreased. After 10 polishing times, the highest unlock success rate was 86.7%. However, after 30 polishing times, the lowest unlock success rate dropped to 24.4%, with the highest one reaching only 35.6%. This decline was attributed to the polishing process, which resulted in shallower fingerprint ridges, making fingerprint recognition more challenging.

The above test results indicated that preparing fingerprint membranes through fingerprint impression is susceptible to the degree of wear of the fingerprint. The greater the wear of the fingerprint, the lower the unlock success rate of the obtained fingerprint membrane. Wear and tear on fingerprints leads to a decrease in the unlock success rate when unlocking a smartphone fingerprint lock, causing inconvenience in daily life. Therefore, there are significant limitations to using the fingerprint impression method for fingerprint backup.

### 4.2. Impact of Finger Wear on the Unlocking Capability of Fingerprint Membranes Produced by SLA Printing

The SLA printing technology, a form of digital manufacturing, allows for preprocessing two-dimensional fingerprint images and subsequently creating three-dimensional fingerprint models. This ensures that the final fingerprint membrane is not influenced by the wear of the fingerprints. In the preparation of fingerprint membranes using SLA printing, factors such as image processing methods, ridge height of the three-dimensional model, and printing process parameters are crucial in determining the final outcome. In this experiment, the recommended process parameters for the employed equipment (100.0% infill, layer height of 30 μm) were used for printing. Therefore, this section further investigated the impact of factors such as fingerprint image processing and ridge height on the effectiveness of SLA-printed fingerprint molds.

#### 4.2.1. Impact of Fingerprint Image Processing on the Unlocking Effect of Fingerprint Films

In order to investigate the impact of fingerprint image processing on fingerprint unlocking efficacy, fingerprints that had undergone 30 rounds of polishing were selected, dividing them into two experimental groups. These groups comprised untreated fingerprint images (Figure 8a) and processed fingerprint images (Figure 8f). Subsequently, three-dimensional fingerprint models were constructed using both sets of images (Figure 8b,g), with ridge height set at 60 μm (Figure 8c,h). Employing the SLA printing technology, we fabricated corresponding fingerprint molds (Figure 8d,i), followed by casting the fingerprint membranes (Figure 8e,j). The examination of Figure 8 reveals that untreated fingerprint images exhibited indistinct lines, resulting in a fingerprint membrane with punctate protrusions. In contrast, processed fingerprint images eliminated powder adherence between fingerprints, yielding internal lines within the fabricated fingerprint mold that were more fluid and distinct. This observation suggests a positive influence of fingerprint image processing on enhancing the quality and precision of fingerprint models.

The detailed results of the fingerprint membrane unlocking tests are presented in Table 1. The experimental outcomes indicated that directly modeling fingerprint membranes using untreated fingerprint images resulted in subpar unlocking performance, with unlocking success rates consistently below 30%, reaching a minimum of 20%. Conversely, the use of processed fingerprint images significantly enhanced the unlocking capability of the fabricated fingerprint membranes, achieving success rates exceeding 60.0%. A comparative analysis with Figure 7 revealed that the unlocking performance of fingerprint membranes produced using SLA printing without image processing was even lower than that of fingerprint membranes obtained through the fingerprint impression method. This disparity may be attributed to the “powder dusting” method employed by SLA printing, which is prone to fingerprint line and feature point omissions, as well as to powder adhesion issues, consequently diminishing the unlocking success rate of the fingerprint membranes. The removal of powder adhesion from the fingerprint images using Photoshop CS6 software, coupled with the application of fingerprint image enhancement algorithms to address features such as “blank spaces” and “breaks”, proved effective for detecting finer details of the fingerprints. This, in turn, enhanced the fingerprint lock’s ability to identify and validate fingerprint characteristics, thereby elevating the overall unlocking success rate.

#### 4.2.2. Influence of Different Ridge Heights on the Unlocking Effect of Fingerprint Films

To analyze the impact of ridge height settings in fingerprint modeling on the unlocking efficacy of fingerprint membranes, three-dimensional fingerprint models with ridge heights of 30 μm, 60 μm, 90 μm, and 120 μm were established. Printing and inverse molding of fingerprint models with varying ridge heights were conducted, as illustrated in Figure 9. As observed in Figure 9, an increase in the ridge height setting resulted in a more pronounced three-dimensional representation of the fingerprint, yielding clearer fingerprint patterns on both the mold and the fingerprint membrane. Specifically, the fingerprint patterns on the 3D model, mold, and fingerprint membrane with a 30 μm ridge height were the shallowest, whereas those with a 120 μm ridge height exhibited the deepest. This correlation indicated that a higher ridge height setting in the three-dimensional fingerprint model enhanced the three-dimensionality of the fingerprint, leading to more distinct fingerprint patterns on both the molded structure and the resulting fingerprint membrane.

The results of the unlocking tests on fingerprint membranes molded from molds with different ridge heights are depicted in Figure 10. The research findings indicated that within the range from 30 to 120 μm for ridge height in the fingerprint models, all molded fingerprint membranes exhibited unlocking functionality. However, a deeper ridge height does not necessarily translate to superior unlocking performance. 

Specifically, when the ridge height in the fingerprint model was set at 30 μm, the resulting fingerprint membranes exhibited the poorest unlocking capability, with success rates consistently below 26.7%. As the ridge height increased to 60 μm and 90 μm in the fingerprint model, the unlocking success rates of the molded fingerprint membranes correspondingly improved. Notably, when the ridge height in the fingerprint model was set at 90 μm, the obtained fingerprint membranes demonstrated the optimal unlocking performance, with success rates exceeding 84.4% and reaching a value as high as 91.1%. However, when the ridge height in the fingerprint model was set to 120 μm, the unlocking success rates of the molded fingerprint membranes decreased. This phenomenon may be attributed to the fact that the average ridge height of human fingerprints is approximately 60 μm. The closer the ridge height of the 3D-printed backup fingerprint membrane aligns with the actual ridge height of the fingerprint, the higher the unlocking success rate. Conversely, when there is a significant deviation of the backup fingerprint membrane from the actual ridge height, the unlocking success rate decreases [22]. During the fingerprint membrane molding process, a thin layer of Vaseline was applied as a separating agent between the mold and the fingerprint membrane. Additionally, SLA printing involves a certain degree of model shrinkage. Therefore, it is advisable to set the ridge height in the fingerprint model slightly above 60 μm. When set at 90 μm, the molded fingerprint membrane’s ridge height closely approximated the actual fingerprint ridge height of 60 μm, resulting in the highest unlocking success rate.

In conclusion, the fingerprint impression method, which involves the direct imprinting of finger fingerprints, effectively preserved the original fingerprints. However, this method is highly susceptible to the condition of the fingerprint, making it challenging to produce usable fingerprint membranes for users with unclear or damaged fingerprints. In contrast, SLA printing could mitigate the impact of fingerprint wear through image processing and ridge height adjustments. In the future, the 3D printing technology for backing up fingerprints holds the potential for application in prosthetics for individuals with disabilities, providing users with hand disabilities an opportunity to reexperience the convenience of the fingerprint recognition technology. Nevertheless, it is crucial to acknowledge and address potential security concerns associated with this technology in its application [26].

For future research and practical implementation of the proposed fingerprint backup method, it is imperative to address security concerns comprehensively. One possible approach to enhance the security of the backup method is through the implementation of robust encryption techniques. Encrypting stored fingerprint data can prevent unauthorized access and safeguard sensitive information from potential breaches. Additionally, implementing multi-factor authentication protocols, such as combining fingerprint authentication with a secondary verification method like a PIN or token, can add an extra layer of security to the system. By integrating these security measures into the design and implementation of the fingerprint backup method, organizations can mitigate potential risks and enhance the overall security posture of biometric authentication systems.

## 5. Conclusions

This study provides an in-depth analysis of the fingerprint membrane preparation efficacy of two fingerprint backup methods, namely, fingerprint impression and SLA printing, when applied to fingerprints with varying degrees of wear. By delving into the optimization of the SLA process for preparing fingerprint membranes, the study aimed to enhance the quality and performance of fingerprint models, ensuring more reliable outcomes in unlocking tests. The findings of this research hold significant guiding implications for the development and improvement of fingerprint backup technologies. The specific conclusions are as follows:(1)Both fingerprint impression and SLA printing methods successfully produced fingerprint membranes capable of unlocking an in-display fingerprint lock. However, the fingerprint membranes created through the fingerprint impression method were susceptible to finger wear. As the degree of finger wear increased, the unlocking success rate of the molded fingerprint membranes decreased. After 30 times of fingerprint polishing, the lowest unlocking success rate achieved using the fingerprint impression method was 24.4%, and the maximum was only 35.6%.(2)Fingerprint membranes prepared using the SLA printing method for worn fingerprints exhibited a lower unlocking performance in their untreated state compared to fingerprint membranes obtained through the fingerprint impression method. The unlocking success rate ranged from 20% to 26.7%. However, through further image processing and optimization of the ridge height parameters, the quality and unlocking success rate of the fingerprint membranes could be significantly improved. The optimal unlocking success rate reached up to 91.1%.

## Figures and Tables

**Figure 1 sensors-24-02627-f001:**
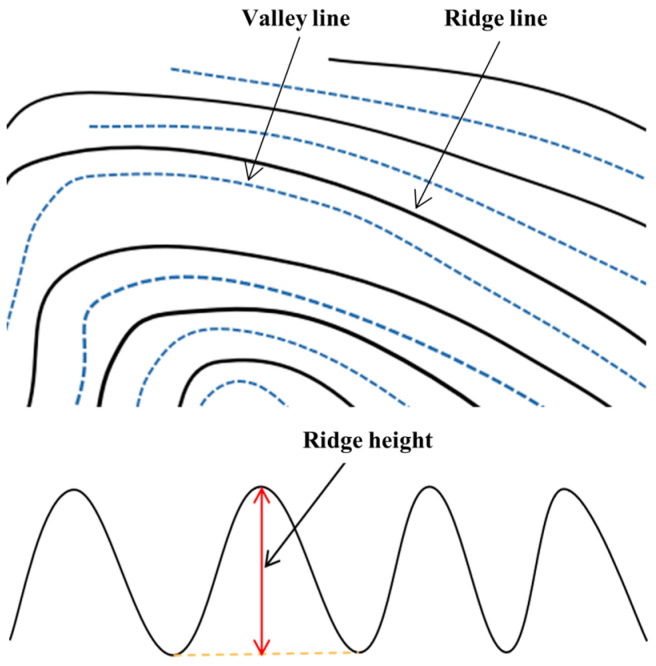
The ridge line, valley line, and ridge height of fingerprints.

**Figure 2 sensors-24-02627-f002:**
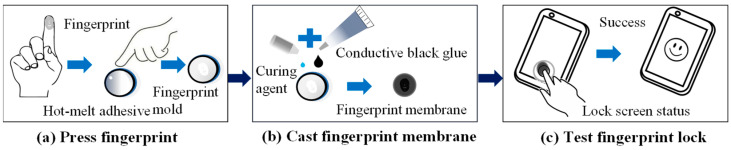
Schematic diagram of the workflow for making fingerprint membranes using the fingerprint impression method.

**Figure 3 sensors-24-02627-f003:**
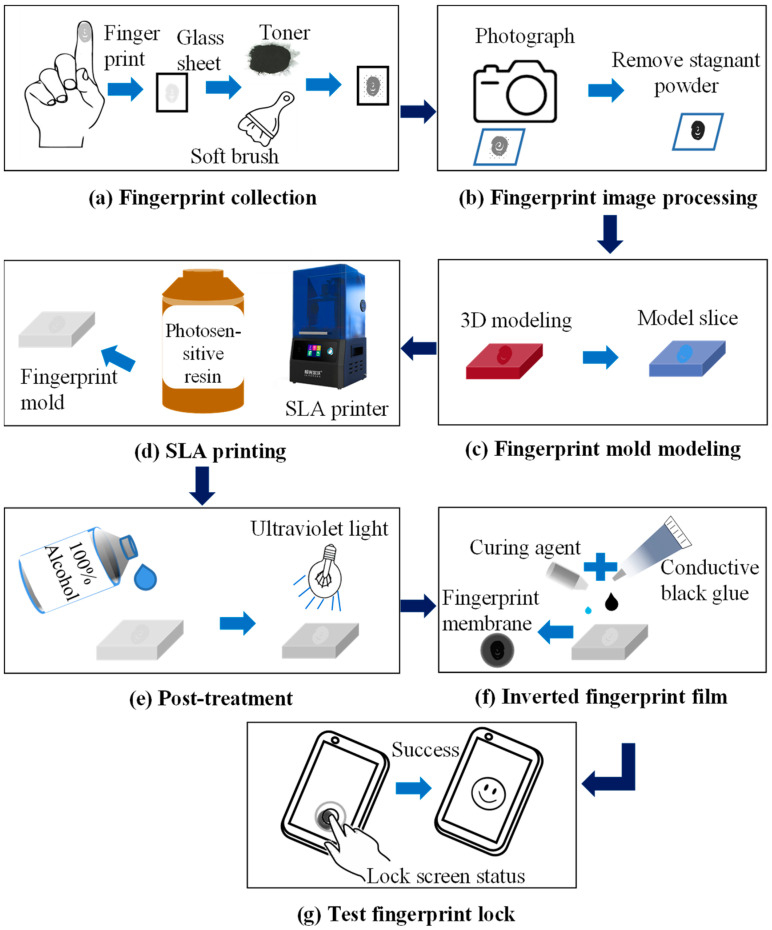
Schematic diagram of the workflow for making fingerprint membranes by the SLA printing method.

**Figure 4 sensors-24-02627-f004:**
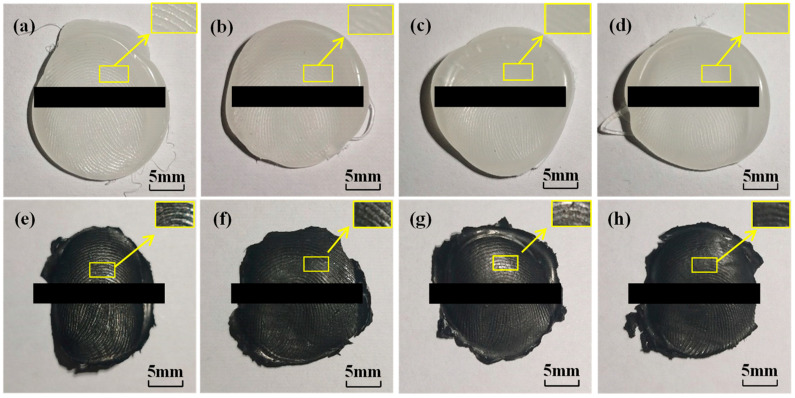
Fingerprint molds and membranes prepared from fingerprints with different degrees of wear using the fingerprint impression method. Polishing times: (**a**,**e**) 0 times, (**b**,**f**) 10 times, (**c**,**g**) 20 times, (**d**,**h**) 30 times.

**Figure 5 sensors-24-02627-f005:**
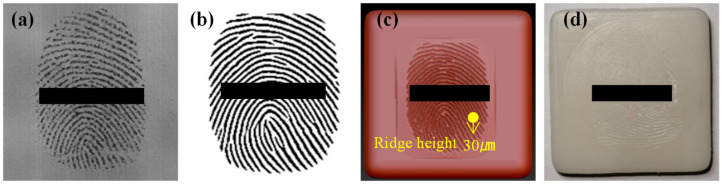
(**a**) Fingerprint image collected through the powder dusting method, (**b**) fingerprint image after processing, (**c**) three-dimensional fingerprint model built by Zbrush 2022 software, (**d**) fingerprint mold printed by SLA. Note: fingerprint ridge height was 30 μm.

**Figure 6 sensors-24-02627-f006:**
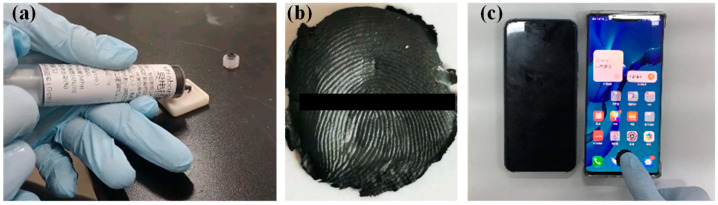
(**a**) Preparation of the fingerprint membrane, (**b**) fingerprint membrane, (**c**) unlocking test.

**Figure 7 sensors-24-02627-f007:**
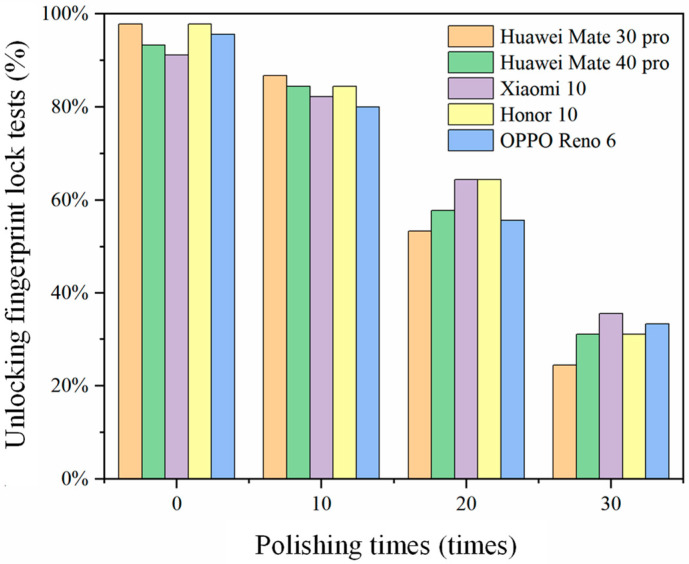
The unlocking pass rate of fingerprint membranes obtained after polishing for a different number of times.

**Figure 8 sensors-24-02627-f008:**
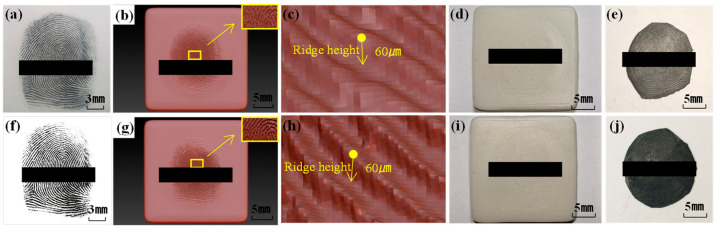
Using (**a**–**e**) unprocessed and (**f**–**j**) processed fingerprint images to backup fingerprints. (**a**,**f**) Fingerprint image, (**b**,**g**) 3D fingerprint model, (**c**,**h**) partial enlarged view of the 3D fingerprint model, (**d**,**i**) printed fingerprint mold, (**e**,**j**) fingerprint membrane.

**Figure 9 sensors-24-02627-f009:**
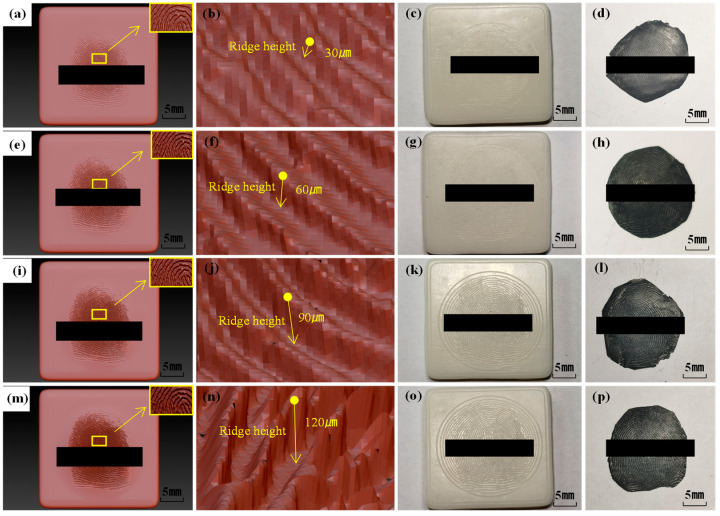
Fingerprint membranes made from three-dimensional models of different ridge heights. (**a**–**d**) 30 μm, (**e**–**h**) 60 μm, (**i**–**l**) 90 μm, (**m**–**p**) 120 μm.

**Figure 10 sensors-24-02627-f010:**
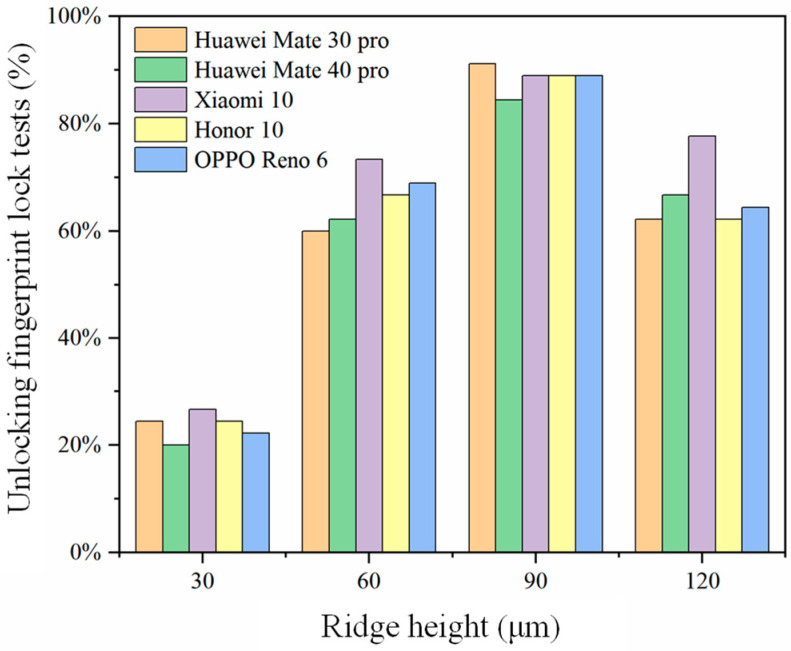
The unlocking pass rate of fingerprint membranes obtained from different fingerprint ridge height models.

**Table 1 sensors-24-02627-t001:** Results of the fingerprint membrane unlocking test.

	Huawei Mate 30 Pro	Huawei Mate 40 Pro	Xiaomi 10	Honor 10	OPPO Reno 6
Untreated fingerprint images	20.0%	28.9%	22.2%	24.4%	26.7%
Processed fingerprint images	60.0%	62.2%	73.3%	66.7%	68.9%

## Data Availability

Data are contained within the article.

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
