# Peer review of "Innovative Solutions for Worn Fingerprints: A Comparative Analysis of Traditional Fingerprint Impression and 3D Printing"

_sensors, 2024, doi:10.3390/s24082627_

Round 1
Reviewer 1 Report
Comments and Suggestions for Authors
The study offers significant insights into fingerprint backup methods, highlighting the potential of 3D printing technology for individuals with worn fingerprints. To enhance its contribution further:
Broaden the Literature Review: Include a wider range of studies on fingerprint recognition technologies to provide deeper context and support for the research.
Standardize Figure Presentation: Adjust Figure 2 for uniform graphical box sizes, and increase the size of Figure 3 to improve detail visibility and overall readability.
Expand Sample Size: Future research should consider a larger sample size to strengthen the findings' generalizability.
Include Biometric Comparisons: A comparison with other biometric technologies could offer valuable insights into the advantages and limitations of fingerprint backups for future works.
These recommendations aim to refine and extend the study's impact within the biometric authentication field.
Reviewer 2 Report
Comments and Suggestions for Authors
This research is a very important step in the development of fingerprint recognition systems. The authors of the study present a comparative analysis of two fingerprint backup methods - the traditional fingerprint and the use of 3D printing technology. The study results showed that the traditional method has high accuracy in reproducing fingerprint entrainment, achieving an impressive unlocking success rate of 97.8%. However, when dealing with worn fingerprints, this method has a reduced success rate, which decreases as the degree of wear increases. In contrast, 3D fingerprint printing, with edge height optimization, reduces the impact of fingerprint wear on unlockability, demonstrating unlock success rates of 84.4% or higher. Therefore, the use of 3D printing technology represents a significant benefit for individuals with severely worn or incomplete fingerprints. The article provides interesting fundamental results but some details should be improved.
1. Please, correct ref. 1 and 2 in line 30 and 31 with follow format [1] and [2].
2. Fig. 7 and 10. I suppose that one test for unlocking was done several times. Please add information how many times was repeated one unlock test and add error bars for fig. 7 and 10.
3. How durable and reproducible such fingerprint backups?
4. It should be noted that this study did not conduct a sufficiently in-depth analysis of the security of using these fingerprint backup methods. An important aspect when using such technologies is protecting data from unauthorized access. Could you add in conclusion part an information of possible ways of security of new proposed fingerprint backup method.
Despite this, the study represents a valuable contribution to the field of biometric technology and deserves attention as a potential solution to the problem of worn fingerprints. Such work may be allowed to be published, but with corrections and suggested improvements.
Reviewer 3 Report
Comments and Suggestions for Authors
I have added the comments and suggestions in the attached file.

Round 2
Reviewer 3 Report
Comments and Suggestions for Authors
The authors answered the questions one by one. The manuscript should be accepted after the authors carefully check the grammar.
Author Response
Thank you for your advice, we have asked colleague who speaks English to correct the details in this manuscript and highlighted the modifications. Furthermore, we have checked using Grammarly and "Grammar Check" to ensure there are no grammar issues.